# Examining health sector stakeholder perceptions on the efficiency of county health systems in Kenya

Lizah Nyawira[1], Rahab Mbau[1], Julie Jemutai[1], Anita Musiega[1], Kara Hanson[2], Sassy Molyneux[3,4], Charles Normand[5], Benjamin Tsofa[3], Isabela Maina[6], Andrew Mulwa[7], Edwine Barasa[1,4]*

1 Health Economics Research Unit, KEMRI-Wellcome Trust Research Programme, Nairobi, Kenya, 2 Faculty of Public Health and Policy, London School of Hygiene and Tropical Medicine, London, United Kingdom, 3 Health Systems and Research Ethics Department, KEMRI-Wellcome Trust Research Programme, Kilifi, Kenya, 4 Centre for Tropical Medicine and Global Health, Nuffield department of Medicine, University of Oxford, Oxford, United Kingdom, 5 Centre for Health Policy and Management, Trinity College, the University of Dublin, Dublin, Ireland, 6 Health Financing Department, Ministry of Health, Kenya, 7 County department of health, Makueni county government, Kenya

* ebarasa@kemri-wellcome.org

**Data Availability Statement:** All the data used in this analysis is provided in the manuscript.

**Funding:** This work was funded by a MRC/FCDO/ESRC/Wellcome Trust Health Systems Research

## Abstract

Efficiency gains is a potential strategy to expand Kenya's fiscal space for health. We explored health sector stakeholders' understanding of efficiency and their perceptions of the factors that influence the efficiency of county health systems in Kenya. We conducted a qualitative cross-sectional study and collected data using three focus group discussions during a stakeholder engagement workshop. Workshop participants included health sector stakeholders from the national ministry of health and 10 (out 47) county health departments, and non-state actors in Kenya. A total of 25 health sector stakeholders participated. We analysed data using a thematic approach. Health sector stakeholders indicated the need for the outputs and outcomes of a health system to be aligned to community health needs. They felt that both hardware aspects of the system (such as the financial resources, infrastructure, human resources for health) and software aspects of the system (such as health sector policies, public finance management systems, actor relationships) should be considered as inputs in the analysis of county health system efficiency. They also felt that while traditional indicators of health system performance such as intervention coverage or outcomes for infectious diseases, and reproductive, maternal, neonatal and child health are still relevant, emerging epidemiological trends such as an increase in the burden of non-communicable diseases should also be considered. The stakeholders identified public finance management, human resources for health, political interests, corruption, management capacity, and poor coordination as factors that influence the efficiency of county health systems. An in-depth examination of the factors that influence the efficiency of county health systems could illuminate potential policy levers for generating efficiency gains. Mixed methods approaches could facilitate the study of both hardware and software factors that are considered inputs, outputs or factors that influence health system efficiency. County health system efficiency in Kenya could be enhanced by improving the timeliness of financial flows to counties and

Initiative (HSRI) grant no MR/R01373X/1 awarded to EB, JJ,KH,BT,CN,SM. Additional funds from a Wellcome Trust core grant awarded to the KEMRI-Wellcome Trust Research Program (#092654) supported this work. The funders had no role in study design, data collection and analysis, decision to publish, or preparation of the manuscript.

**Competing interests:** The authors have declared that no competing interests exist.

health facilities, giving health facilities financial autonomy, improving the number, skill mix, and motivation of healthcare staff, managing political interests, enhancing anticorruption strategies, strengthening management capacity and coordination in the health sector.

## Introduction

Kenya has made a commitment to achieve Universal Health Coverage (UHC) by the year 2030 [1]. The country's UHC policy outlines 4 objectives namely a) strengthen access to health services b) ensure quality of health services, c) protect individuals and households from the financial risks of ill health d) strengthen the responsiveness of the health system. The country implemented a UHC pilot in selected (4 out of 47) counties that entailed the removal of user fees in public hospitals and is currently scaling up the UHC programme in the form of provision of health insurance subsidies to poor households. Despite Kenya's commitment to achieve universal health coverage (UHC), this aspiration faces, among others, the challenge of a constrained fiscal space for health [2, 3]. For instance, Kenya's public expenditure on health is 2.3% of the country's gross domestic product (GDP) [3], far lower than the recommended level of 5% required to achieve UHC [4]. Improving the efficiency of health systems is one of the key strategies for unlocking additional resources in the health sector [5, 6], needed to advance the country's UHC goal.

Health system efficiency refers to the extent to which health system objectives are met given the resources invested in the system [7]. Two types of efficiency, technical and allocative efficiency, have been distinguished [8]. Technical efficiency is achieved when resources are allocated such that outputs are maximized for a given level of inputs, or inputs are minimized for a given level of outputs [9]. Allocative efficiency is achieved when resources are allocated such that outputs are maximized for a given level of input cost, or input costs are minimized for a given level of outputs [9]. Given the scarcity of healthcare resources, it is imperative that health systems orient their operations towards using their resources efficiently to optimize the achievement of stated health system goals. It has been estimated that 20% to 40% of health system spending globally is wasted through inefficiency [10]. Efficiency measurement is therefore a key dimension of health system performance assessment.

Kenya's healthcare system is pluralistic, with service provision provided by both public and private healthcare facilities in almost equal measure. The public healthcare delivery system is organised into four tiers, namely community (comprising of community units), primary care (comprising of dispensaries and health centers), county referral (comprising of first and second referral hospitals) and national referral (comprising of tertiary care hospitals) [11]. The health system is financed by revenues collected by [12]: a) The government (national and county) through taxes and donor funding, b) The National Hospital Insurance Fund (NHIF) through member contributions, c) Private health insurance companies through member contributions, and d) Out of pocket spending by citizens at points of care. Purchasing of healthcare services is carried out through: (a) Supply-side subsidies to public facilities by national and county governments; for instance, the county departments of health provides budgets to county hospitals to finance service delivery to citizens within the county, (b) The NHIF, which contracts public and private healthcare facilities in Kenya and pays them for services provided to its enrolled members, and (c) Private health insurance companies that contract private healthcare facilities and pays them for services provided to their enrolled members [13]. The Kenyan health system is dependent on donor funding and out of pocket payments, with the

two contributing 19.1% and 23.3% of total health expenditure, respectively according to the most recent national health accounts [14].

In parallel with Kenya's UHC push, the country devolved its governance arrangements in 2013, with the formation of two tiers of government: a national government and 47 semi-autonomous county governments [15]. Within the health sector, decentralization, and more specifically devolution entailed the transfer of ownership and management of county health-care facilities (county hospitals, health centers and dispensaries) and healthcare service delivery to the county level, while the central Ministry of Health retained the management of national referral hospitals, health policy and regulatory functions [16, 17]. Kenyan counties receive block grants from the central government, and in addition collect revenues locally, and have absolute control over their budgets and priorities [15]. They allocate funds to service areas and units, including public health facilities, based on their priorities. The relative performance of county health systems can therefore be attributed to their capacity to efficiently and effectively allocate and use available resources.

County health systems are critical determinants of overall health system efficiency in Kenya given their central role in service provision and significant resource consumption. For instance, counties consumed 60% of the total government budget for health in the fiscal year 2015–2016 [18]. Decentralization, of which devolution is a specific form, has been promoted as a key reform for improving health service delivery, among others, improving health system efficiency [19, 20].

Efficiency analysis is increasingly carried out in healthcare, but most of these studies analyse the efficiency of healthcare organizations (such as hospitals and health centres) [21, 22]. Few studies examine the efficiency of national and sub-national health care systems [21, 22]. It has been argued that empirical measurement of efficiency at system level could be useful for health system decision makers and managers [22].

Understanding the perceptions of health sector stakeholders on efficiency and the factors that influence it is a useful initial step in efficiency analysis. This is because health sector stakeholders have knowhow from their lived experiences, reflecting different perspectives on health systems, and could provide insights that can inform the formulation and refinement of relevant research questions for the quantitative assessment of health system efficiency. This paper presents research that is part of a larger study which aims to examine the level and determinants of the efficiency of county health systems in Kenya. In this paper we present findings from the analysis of group discussions of health sector stakeholders in Kenya on their perceptions of how efficiency of county health systems in Kenya can be conceptualized, and the factors that influence the efficiency of county health systems.

## Methods

### Study design and data collection

We used a qualitative cross-sectional design [23]. We chose a qualitative approach to facilitate the identification of factors that influence the efficiency of health systems that are not easily captured quantitatively ("soft" factors) and provide a starting point for further work to develop approaches to quantify these factors, where feasible, and explore their effect on health system efficiency. Qualitative methods also enrich efficiency analysis by facilitating an examination of the mechanisms of the relationships between health system efficiency and its determinants (i.e. it allows for the examination of the "how's" and "why's" of this relationship). Understanding these mechanisms provides evidence that is richer than mere identification of determinants of efficiency and can potentially inform policy design to improve health system efficiency.

We adopted pragmatism as an interpretative framework, placing importance on the problem being studied, the questions asked, and the practical implications of the research [23, 24]. Drawing on this interpretive framework, the study was guided by the ontological belief that reality is what is useful and practical, and the epistemological belief that reality is known using multiple objective and subjective approaches [23, 24].

We collected data using focus group discussions. We chose to use the focus group discussion approach because we are interested in collective views of stakeholder and because the method is useful in generating a rich understanding of participant experiences [25]. The focus group discussions were conducted in a one-day stakeholder workshop we organized in April 2019 to deliberate on the efficiency of county health systems in Kenya. We drew workshop participants from policy makers and health system managers at the national level (Ministry of Health, academia, and development partners), and at the county level. The objective of the workshop was to engage health sector stakeholders to obtain their views about the factors that should be investigated to understand if and how they influenced the efficiency of county health systems. For this study, we adopted purposive sampling of participants to gain diverse perspectives from respondents in order to investigate our research objectives. This included representatives from 1) National Ministry of Health's Monitoring and Evaluation, Policy and Planning, and Health Financing units, 2) the multi-sectoral Monitoring and Evaluation technical working group, 3) the multi-sectoral Health Financing technical working group, 4) Development Partners for Health in Kenya (DPHK), a forum for local and international donor organizations that support the Kenyan health sector, and 5) participants from county health departments representing 10 out of the 47 counties in Kenya. A total of 25 workshop participants from diverse backgrounds were selected as detailed in Table 1.

In the planning stage of the workshop, the participants were approached through email that detailed the aim of the study and invited them to participate in the workshop. This was then followed up by phone calls to confirm attendance. The national Ministry of Health (MOH) and all the 10 county governments that were requested to send participants did so, signalling their keen interest in the study. Three of the researchers, RM, BT, and EB, had prior professional interactions with the national level participants while working on other research projects. None of the researcher had prior interactions with the majority of the participants who were drawn from counties. All participants were aware about the goals of the research project and the role of the workshop as the formative phase of a larger study on health system efficiency. The workshop was structured into three parts. In the first part we introduced the study and objectives of the workshop. In the second part we divided the participants into 3 groups and facilitated a focus group discussion within the three groups to elicit their views about the factors that influenced the efficiency of county health systems in their settings. Finally, we had a feedback session where each group shared summary points that were then discussed by the entire group. We obtained verbal consent to audio record the proceedings of the discussions. We supplemented the audio recordings with note taking. Each of the focus group discussions was facilitated by co-investigators in the study; Group 1 led by CN (n = 5), Group 2 led by EB (n = 6) and Group 3

**Table 1. Number of study participants.**

| Category of participants | Male | Female | Total |
|---|---|---|---|
| Ministry of Health | 3 | 4 | 7 |
| County Health Department | 9 | 3 | **12** |
| Development Partners | 2 | 3 | **5** |
| Council of Governors | 1 | - | **1** |
| **Total** | **15** | **10** | **25** |

led by KH (n = 5) assisted by co-authors JJ, SM and RM. All researchers had masters or PhD level training in health economics. LN, AM, RM, BT, EB had primary training as health care professionals. We used semi-structured topic guides to facilitate the discussions (S1 Appendix).

## Data analysis

The audio recordings were transcribed to MS Word by a commercial transcription services providers. LN repeatedly listened to the audio recordings to check the accuracy of the transcription prior to importing the transcribed data to NVIVO version 10 for coding and analysis. We used a thematic analysis to analyse the data (Ritchie and Spencer 1994). LN and EB began by familiarizing with the data by reading the transcripts several times. LN and EB independently developed an initial coding framework based on the questions used to facilitate the discussions. LN and EB then compared and harmonized their coding frameworks and used insights from the anonymised transcribed data to refine and modify the coding framework. LN then applied the refined coding framework to code the transcripts. LN subsequently charted the data and categorized them into themes. Finally, LN and EB interpreted the data by identifying connections between the various themes and using this to gain a better understanding of participant perceptions about the factors that influence the efficiency of county health systems. We employed two key approaches to enhance the rigor and trustworthiness of the study findings. First, carried out member checking [26] by sharing preliminary findings with selected participants to check that the findings are a true reflection of the discussions. Second, we carried out peer debriefing [26], where we shared the preliminary findings with a group of other researchers in Kenya to obtain their disinterested views about findings.

## Ethics

This study received ethics approval from the KEMRI Scientific and Ethics Review Unit (SERU), approval number KEMRI/RES/7/3/1. All participants were provided with written briefing notes and an oral briefing providing the study details, procedures, and intention to analyse and publish the findings from the focus group discussions. Participants were requested to provide verbal informed consent after this briefing. Specifically, they provided informed verbal consent for the sessions to be audio recorded, analysed and presented as part of this study. Verbal consent was preferred by participants to enhance the confidentiality of the discussions. We ensured confidentiality by anonymizing the transcribed data using codes, and labelling quotes with respondent categories rather than respondent names. We restricted access to transcribed data to research team members.

# Results

In this section, we present study findings organized in x thematic areas namely stakeholder understanding of the efficiency of county health systems, relevant inputs to the county health system, relevant outputs of the county health system, and factors affecting county health system efficiency.

## Stakeholder understanding of the efficiency of county health systems

Stakeholders generally understood efficiency to mean the best use of available resources to optimize desired health system outcomes. They saw the county health system as utilizing health system inputs to produce health system outcomes and regarded efficient county health systems as those that optimize this process to maximize health system outcomes. Health system inputs are those resources that are utilized by the health system to achieve its objectives. Outputs on the other hand

are the intermediate or final intended goals of the health system. Participants highlighted that efficient healthcare delivery should be cost effective and responsive to community healthcare needs. Therefore, a system wide approach might be needed for improving health system efficiency.

> *"I think it is about maximizing our outputs and trying to get the best we can from the little inputs we have. That is what we call efficiency" (Development partner 1, FGD 3)*

> *"Efficiency for me is how our systems are working to make health care delivery less costly and more responsive to the community needs. For example, what systems do we have? Are they responsive? If it's the governance issues, are they responsive to our systems needs as well as the community needs? How is every other system interacting to like facilitate health care delivery" (MoH Official 3, FGD 2)*

Stakeholders pointed out that the process of transforming health system inputs into desired outcomes is affected by factors within and external to the health sector. In some cases, the formal county health authority may not have practical authority for health system resource allocation & distribution in the country. It was indicated that understanding the efficiency of county health system required an understanding of these factors.

> *"I find it very complex to determine how efficient a health system is because the health sector has so many factors that have to come in play to produce something. For example, there are some factors at the community level that will determine how efficient resources will be used at the facility level, the management at the county level, leadership at the county level" (MoH Official 2, FGD 3)*

> *"It's beyond the department of health because there are so many other factors and decision makers who have veto power over resource management and have a significant influence on the efficiency of county health systems. For example, at the county level, it is always assumed that the County executive for health (CEC) has the overall authority over the department of health and its resources but in practice that's not necessarily the case." (Researcher 2, FGD 3)*

## Relevant inputs to the county health system

The optimal use of healthcare inputs determines the efficiency of health systems. Workshop participants distinguished between "hard" and "soft" inputs to the county health system production process. Hard inputs were consistent with health system strengthening components and included county financial resources, human resources for health, health sector infrastructure, and healthcare commodities (e.g. medicines). Soft inputs included less tangible resources such as governance systems, policies and guidelines, managerial systems, and the relationships between stakeholders. Participants highlighted the need for emphasis on soft inputs and also expressed the difficulty in assessing them to ascertain the health system efficiency.

> *"Can I call them hard inputs? the ones we can touch for example staff, commodities, infrastructure, equipment. Then there are the soft inputs such as decision-making skills and how well resources are managed. It is easy to identify and measure hard inputs, but difficult to measure the soft inputs"* (MoH Official 2, FGD 3)

> *"Several inputs are important; people, drugs, facilities, transport etc. However, software which we know is a really important part of health systems is also important. This includes managerial practices, relationships, politics." (Researcher 3, Joint Forum)*

Workshop participants highlighted the importance of having the optimal mix of inputs for county health system efficiency, and the need for good coordination between the various inputs.

> "If you look at the counties, we have so many health workers and few resources for operations. There is no need of sending a neurosurgeon to the county when in the first place you don't even have a theatre for them. So, as we plan, we must look at bringing all the pillars of the health system together. There is always a disconnect when you have so many gardeners but no tools to work with, or you have too many tools and no gardeners" (County Official 1, FGD 2)

> "We had an interesting case where we had a urologist and we didn't even have urology towers. We were just lucky that he was kind and would bring his private equipment to work in a government hospital. If we bring in a urologist, we should give them resources to work with." (County official 1, FGD 2)

## Relevant outputs of the county health system

The level of health output in relation to the level of inputs, the quality of outputs, and link between the outputs and health system goals determine the efficiency of health systems. Workshop participants felt that the health output & outcome indicators used in quantitative assessment of county health system efficiency should represent the disease burden of the county. They also highlighted the need to select standardized outputs and outcomes indicators that could be implemented nationally across all counties of Kenya to facilitate valid comparisons. Participants highlighted the dual burden of disease in Kenya, with an increasing burden of NCDs along with a continued relevance for infectious diseases & reproductive, maternal, newborn, child and adolescent health (RMNCAH). Therefore, respondents expressed the need for monitoring of intervention coverage and health outcome indicators relevant for both NCDs and infectious diseases which could be compared across 47 counties.

> "For outputs, I would look at things like per capita utilization and still go for the traditional indicators like the RMNCAH indicators for comparability across the 47 counties." (MoH official 2, FGD 3)

> "Looking at the health indicators by using our local health indicators, we should be thinking about NCDs. But we are still talking about infectious disease indicators which don't work for some counties. . .Let's develop tools that don't just look at the conventional health indicators that are meant for the poor because different counties have different issues." (County Official 1, Joint Forum)

In addition to indicators of intervention coverage and health outcomes, participants highlighted the relevance of including quality of care provided at the healthcare facilities as an important dimension for assessing the health system efficiency.

> "I think we don't just want to talk about numbers for example the proportion delivered. We want to talk of quality. If you look at the constitution, it talks about the right to the highest standard of health. So, when we talk of outcomes, we must consider quality." (MoH official 2, FGD 3)

> "We need not just to talk about the numbers, we also need to take into consideration the quality of care that is offered" (County Official 1, FGD 2)

## Factors affecting county health system efficiency

**Public finance management.** Workshop participants identified several aspects of public finance management (PFM) in the health sector as influencing the efficiency of county health systems. One of this was delays in disbursement of funds from the national level to the county level, and from the county level to the facility level. The smooth flow of funds without unnecessary delays in disbursements resulted in more efficient county health systems.

"*I will give an example of UHC pilot. Money was received by counties by 20[th] December [2018] and three to four months down the line, money was still not at facility level and these are facilities that are not collecting user fees. So, you can imagine their operations.*" (MoH Official 4, FGD 2)

A second aspect of PFM that influenced efficiency of county health systems was financial autonomy of health facilities. Participants reported that the practice in most counties, in which health facility revenues (user fee collections, National Hospital Insurance Fund reimbursements, and budget allocations by the national and county governments) are redirected to a central county account (the county revenue fund) removed the financial autonomy of public health facilities and negatively affected their operations.

"*Is there facility autonomy? Because if the money is meant for health services but not all of it is going to health care, that brings about inefficiency. Would it be better to have policies that will devolve those resources further, so that at the county level, if this money is allocated to hospital A or a dispensary B or a certain community unit you cannot divert it to another cause*" (MoH official 6, FGD 1)

"*We have a challenge in most of the counties whereby the financial collection from facilities are all directed to the County Revenue fund (including the FIF). And when facilities need money it is a bottle neck to the funds being available down at the facilities.*" (County official 7, FGD 2)

**Human resources for health.** Participants also identified human resource management as a factor that influenced the efficiency of county health systems. For example, participants felt that the efficiency of county health systems was affected by high workload and inadequate number of healthcare staff employed by the counties.

"*And then also if you are talking of the county efficiency, most likely we are going to the county department of health and looking at how many health workers we have, maybe by cadre, whichever cadre, cross cutting.*" (MoH Official 2, FGD 3)

"*You may find some health centres have only one nurse working, therefore though they are doing data keeping, sometimes they scribble today, tomorrow they don't. So, for you to know the exact number of children they have immunized, you may not be able to get an exact figure. They try as much as possible but sometimes they are understaffed.*" (County official 2, FGD 3)

In addition to numbers they also highlighted the importance of having the right skill mix of healthcare staff in the county health systems. There was maldistribution of healthcare staff, especially medical specialists, across the counties. This was due to absence of any joint formal recruitment strategy and lack of cooperation for sharing healthcare staff between counties. Participants suggested a coordinated sector wide collaboration and sharing of resources for an improved health system efficiency.

*"I am told of a county that does not have medical officers but has specialists. They deploy specialists to health centres because they do not have medical officers and they have more than enough patients. . .. How can we as counties optimize the human resources that we have? For example, county A can have five-ten surgeons in one sub-county hospital whereas County B has none. Those are system inefficiencies." (CoG representative, Joint Forum)*

*"We were told that a certain surgeon works in three Counties. If he can work and see all these patients, why can't the three Counties pay that one surgeon? Maybe there aren't many surgical cases in one county. If we can talk as a sector, we would achieve more efficiency." (MoH Official 3, FGD 2)*

*"I think moving forward, sharing of resources among the counties is going to be important because there are some places where there is oversupply of some human resources, yet it is difficult to find a smooth mechanism where those resources can be shared with other counties that may not have. And if we were to find a way of sharing staff, especially as you go up with more specialized resources, it may improve efficiency."* (Researcher 1, FGD 1)

Participants reported that challenges in the way counties manage their healthcare staff had resulted in reduced staff motivation which in turn negatively affected the efficiency of county health systems. These challenges included delays in payment of salaries, inadequate structures for staff promotions and transfers, and poor resourcing of health facilities. For example, counties had experienced frequent healthcare staff strikes that disrupted health service delivery. Further, the level of absenteeism of healthcare staff was reported to be high. Moreover, there are also governance issues where the officials do not consider healthcare staff as a priority, resulting in delayed payment of salaries.

*"There was a service delivery survey by World Bank, and one of the major things highlighted was that, we may have 'human resource' but the level of absenteeism was quite high in some counties." (MoH Official 2, Joint Forum)*

*"Governors do not pay their staff on time for no apparent reason, yet they have already received the resources from the national government. They think that staff in healthcare are not a priority and we have seen a lot of industrial unrest in the sector." (MoH Official 3, FGD 2)*

Participants also reported that inadequate accountability mechanisms for especially permanent & pensionable healthcare staff contributed to health system inefficiency. One source of poor accountability was the absence of an effective staff performance appraisal system. Another was that healthcare staff in the public sector had permanent employment contracts.

*"And in terms of even the outcomes or outputs expected, most of our facilities stopped the appraisal system. You know the government before used to enforce appraisal for all staff, now it's done if the county feels like." (County official 1, FGD 2)*

*"One source of inefficiency is the way health workers are managed. Health workers are not held accountable because they have permanent and pensionable employment terms. You will find a surgeon who decides he will work for one day in a month. Another one will decide that they are not going to work for the next 3 months and yet nothing can be done to them because they have permanent and pensionable terms" (MoH Official 3, FGD 2)*

**Political interests and interference.**   Participants noted that political interests influenced the efficiency of county health systems by influencing the allocation of health budgets.

Specifically, the local politicians preferred allocating health budgets to tangible capital assets and infrastructure over other forms of intangible investments such as health commodities or health workers. This was because capital assets and infrastructure were more visible and gained the politicians political mileage.

*"What matters to most politicians is things that can be seen. . . a boat, a big building, infrastructure, health facilities everywhere, even when you really don't need them. . .that is why we have so many white elephants around, because people have put things which are not necessary. Politicians seem to win the day when it comes to health. . .." (County official 1, FGD 2)*

*"Political interference affects efficiency. . .as a county health manager, you cannot make decisions out of your own experience or your position because you have to be in favor of a certain political leader" (County official 5, FGD 1)*

Participants reported that politicians at both the national and county level used their power to interfere with and influence the allocation of county resources in ways that were not optimal. This resulted in discouraging people from taking up the post of health manager leading to further health system inefficiencies.

*"Political interference by both national and county politicians results in the allocation of resource based on political interests rather than population health needs. An example I can give is the medical equipment service (MES) program where counties are required to spend a specified amount of their development budget to lease medical equipment. We [counties] are spending a big part of our budget on this program. This program was decided and introduced by politicians at the national level without regard of the priorities of individual counties" (County official 5, FGD 1)*

*"In [county x] we built a 150-bed hospital at a cost of KES 140 million. Compare that with the KES 200 million we are required to pay annually for the lease of the medical equipment program. This is what happens when allocation decisions are influenced by politicians rather than technical staff" (County official 7, FGD 2)*

*"Nobody wants to be a health manager in the health sector because of political influence. Everybody is scared because every time you get a lot of interference from politicians. This has negatively affected the motivation of health facility managers. You will find local politicians demanding that certain patients are prioritized over others. Local politicians also interfere with staff recruitment in facilities and you end up with staff that you either do not need or do not have the skills to do the job" (County Official 1, Joint Forum)*

**Corruption.** It was reported that corruption was one of the factors that influenced the efficiency of county health systems. Among others, corruption influenced procurement decisions resulting in counties spending more resources than necessary to purchase healthcare commodities, leading to inefficiency.

*"There is one aspect which is corruption, you may find certain drugs are purchased from outside while another one purchases locally which ends up being more expensive. If you ask why they are purchasing locally you may not get a clear answer. So that is one of the inefficiencies" (County Official 3, FGD 3)*

**Management capacity.** The management capacity of health facility managers was considered to influence the efficiency of county health systems. Respondents felt that the practice by

counties where health workers with clinical backgrounds, but no management training were appointed to management positions, compromised the management of public health facilities.

> *"Another source of inefficiency is poor management of health facilities. It is often assumed that the fact that you are a good doctor means you will be a good manager. The fact that you are good surgeon, you know how to cut [operate] it is decided that you are going to be the manager."* (County Official 1, Joint Forum)

> *"Across our health facilities we have managers that do not have a background in management or administration. You will find that individuals with clinical backgrounds and no management training are picked to oversee health facilities. Clinical officers are picked to manage health centres and medical doctors are picked to manage hospitals. The individuals have no training in financial management or administration and yet you expect them to run health facilities efficiently?"* (County official 7, FGD 2)

**Coordination of actors.** Participants identified inadequate coordination among various health system actors as a source of county inefficiency. They reported that poor coordination between the national ministry of health, and the county departments of health, and between the county departments of health and development partners (donors) led to duplication of efforts.

> *"The national and county government, as well as development partners [donors] are not coordinated in their activities. You will find that two development partners, as well as government are doing the same thing resulting in duplicative allocation of resources. Everyone is doing their own thing and we are so fragmented rather than work in a coordinated way to ensure efficient use of the little resources we have."* (MoH Official 3, FGD 2)

> *"At times we [county governments] don't even know what activities the national ministry of health is doing. We are not aware. It is that bad. You will also find instances where the county governor is not aware of what some development partners are doing in his county. We need to coordinate and work together so that we get to know where and how resources are allocated"* (County official 6, FGD 2)

## Discussion

This study explored healthcare stakeholder perceptions and understanding of the efficiency of county health systems in Kenya. It also examined their views about what factors influence the efficiency of county health systems. While the healthcare stakeholders' understanding of what an efficient health system is aligned with the generally accepted definition of technical efficiency–maximizing health system outputs or outcomes for a given budget-, they recognized the need for efficient health systems to be aligned to population health needs. This resonates with the view that responsiveness to population health needs is one of the key health system goals alongside efficiency, equitable access, financial risk protection and quality [10].

In considering the inputs to be considered in the analysis of health system efficiency, stakeholders highlighted the need to consider both hard and soft inputs. Hard inputs include tangible health system building blocks while soft inputs include less tangible aspects of health systems such as managerial processes, policies and stakeholder relationships. This aligns with the conceptualization of health systems as comprised of both hard and soft elements which is based on the recognition that software aspects of health systems influence their functioning [27, 28]. Software aspects of health systems are, however, rarely included as inputs in the

efficiency analysis of health systems. A review of literature on the efficiency of health systems at the national and sub-national level found that no software aspects of health systems were included as inputs [29]. This is perhaps because it is difficult to capture intangible health systems factors as quantitative variables that can be measured and incorporated in analysis.

Healthcare stakeholders also highlighted the need for health system outputs to be aligned with the changing patterns of disease epidemiology in Kenya. Specifically, they observed that while the country's health system has typically been assessed using measures of intervention coverage and outcomes for communicable disease and reproductive, maternal, neonatal, and child health indicators, there was a need to broaden these indicators to include indicators for non-communicable diseases given that NCDs were emerging as a major source of disease burden in the country. Indeed NCDs now account for 50% of inpatient admission, and 50% of hospital inpatient deaths in Kenya [30]. Stakeholders also indicted the need to consider quality of care as an output of the health system.

Healthcare stakeholders identified several factors they felt influence the efficiency of county health systems in Kenya. First, several aspects of public finance management (PFM) we thought to influence the efficiency of county health systems. These included delays in disbursements of funds from the national government to county governments, and from county governments to healthcare facilities funds. Funding delays affect efficiency by compromising the planning because of the unpredictability of resource availability [31, 32]. Previous studies have documented delays with funding disbursement to counties and health facilities in the Kenyan health system as a challenge [2, 33]. Further studies should examine the reasons for these delays. The fact that public health facilities lacked both financial and procurement autonomy was also identified as a likely source of county health system inefficiency. Reduced autonomy compromised facility managers agency to respond and address emergent issues in the operations of public healthcare facilities and compromised service delivery. Financial autonomy has been identified as one of the PFM factors that impacts the functioning of health facilities in Kenya [34], and the efficiency of health systems in Tanzania and Zambia by imposing budget rigidities that impair health managers' agency to respond to health needs [35]. There is need for follow up studies to unpack other aspects PFM and their influence on county health system efficiency in Kenya.

Second, the number, distribution, motivation, and accountability of human resources for health was thought to influence the efficiency of county health systems. While overall deficiencies in the numbers of healthcare staff have been well documented in Kenya [36], healthcare stakeholders observed that the maldistribution of healthcare staff, such that some counties had more medical specialists than they needed while others had fewer (or none), affected county health system efficiency. The unequal distribution of healthcare staff across regions was shown to contribute to inadequate health system performance in Ghana [37]. Healthcare staff motivation was also identified as a factor that contributes to the inefficiency of county health systems. In Kenya, low staff motivation occasioned by complaints about poor remuneration, inadequate resourcing of the health system, and inadequate capacity of counties to manage the human resource function has manifested in the form of frequent and prolonged health worker strikes [38] and high absenteeism of health workers [39]. Healthcare stakeholders also highlighted poor accountability occasioned by an ineffective performance management and appraisal system as a potential source of inefficiency. Holding healthcare staff answerable for processes and outcome has been identified as a key dimension of human resource for health governance [40]. Inadequate accountability has, for instance, been shown to contribute to healthcare staff absenteeism [41].

Third, political interests were identified as one of the factors influencing the efficiency of county health system by influencing the allocation of resources. Political actors had a

preference for infrastructure investments that were visible since these would gain them political mileage. Political actors hence preferred allocating health budgets to capital assets such as ambulances and medical equipment and building health facilities over other forms of investments such as health commodities or health workers. Political interests have been shown to influence healthcare priority setting, including for commodities and human resources for health in Kenya [42–45] and other settings given that resource allocation is a political process [46].

Fourth, healthcare stakeholders identified corruption as one of the factors that influenced the efficiency of county health systems. Corruption, and especially procurement corruption, led to resource wastage. Corruption has been identified as one of the major causes of resource wastage in health systems [47, 48]. For instance, a study in Kenya reported that resource misallocation and theft compromised HIV service delivery [49] while stakeholders in Nigeria identified various forms of corruption, including procurement-related, informal payments, health financing, and employment-related corruption as compromising the performance of the Nigerian health system [50].

Fifth, inadequate management capacity of health facility managers was thought to affect the efficiency of county health systems. Respondents felt that the practice by counties in which health workers with clinical backgrounds, but no management training were appointed to management positions compromised the management of public health facilities. Management capacity practices has been shown to influence health system performance [51]. For instance, a study in Italy found that managerial competencies are positively associated to organizational performance in the health sector [52]. District and health facility level management has also been shown to be associated with improved health system performance in Ethiopia [53].

Sixth healthcare stakeholders identified the poor coordination between the national MOH, and the county departments of health, and between both national and county departments of health and development partners (donors) led to duplication of efforts and waste of resources. It has been shown that donors influence health sector policies and implementation in low and middle income countries [54]. For instance, fragmented donor approaches was shown to undermine the effectiveness of donor support in Zambia and compromise the implementation of the heath sector strategies [55, 56]. Likewise, the uncoordinated donor support and activities was identified as one of the key sources of inefficiency of the Democratic Republic of Congo health system [57]. In Ghana, poor coordination across ministry of health agencies was shown to result in duplication and reduced clarity or roles, which in turn resulted in inefficiency [58].

Finally, this work highlights the potential contribution of qualitative research in assessing the efficiency of health systems. Efficiency analysis in healthcare is dominated by quantitative analysis using frontier approaches (data envelopment analysis and stochastic frontier analysis) [59]. A literature review carried out by Mbau et al (2021) found that only 3% and 2% of studies that assessed the efficiency of national or sub-national health systems used qualitative or mixed methods approaches, respectively. The use of qualitative approaches to solicit stakeholder views augments these considerations and potentially improves the validity of the selection.

The study has several limitations. First, the study used only one data collection approach (stakeholder discussions). Using multiple data collection methods would have improved the rigor of study by facilitating the triangulation of the data. Second, the focus group discussions that formed the basis for data collection comprised of participants of varied positional seniority. It is likely that junior participants felt constrained from airing their views freely in the presence of their superiors. Lastly, while this is a qualitative study that does not require formal sample size calculation and does not aim for statistical significance, the number and diversity of study participants from each county is small (1–2 per county) limiting the richness of

potential views from each county. These limitations notwithstanding, a strength of the study is the use of thick description of the study context with rich data excerpts that might help in the transferability of study findings into similar care contexts.

## Conclusion

This study reported views of health system stakeholders in Kenya on the efficiency of county health systems. Stakeholders not only shared their understanding of health system efficiency, but also identified factors that they considered to influence the efficiency of county health systems. A key highlight of the findings is the fact that the factors identified included both hardware and software aspects of the system. These include public finance management, human resources for health, political interests, corruption, management capacity, and poor coordination. The implication of these findings is that for the analysis of health system efficiency in Kenya and other settings to be comprehensive, it will need to examine both hardware factors that are easily quantified and software factors that are harder to quantify and incorporate in standard quantitative approaches to efficiency analysis. This means that comprehensive efficiency analysis will need to employ mixed methods that include both quantitative and qualitative approaches. The findings also demonstrate the value of engaging health sector stakeholders to solicit their views to as part of health system analysis such as efficiency analysis. For policy makers, the study findings suggest that the efficiency of county health systems could be enhanced by improving the timeliness of financial flows to counties and health facilities, giving health facilities financial autonomy, improving the number, skill mix, and motivation of healthcare staff, managing political interests, enhancing anticorruption strategies, strengthening management capacity and coordination in the health sector.

## Supporting information

**S1 Appendix. TOPIC guides for focus group discussions.**
(DOCX)

## Acknowledgments

We acknowledge all the health system stakeholders from the national and county level, as well as non-state actors who attended and participated in the stakeholder workshop.

## Author Contributions

**Conceptualization:** Julie Jemutai, Kara Hanson, Sassy Molyneux, Charles Normand, Benjamin Tsofa, Isabela Maina, Andrew Mulwa, Edwine Barasa.

**Data curation:** Rahab Mbau, Julie Jemutai, Anita Musiega, Kara Hanson, Sassy Molyneux, Charles Normand, Benjamin Tsofa, Isabela Maina, Andrew Mulwa, Edwine Barasa.

**Formal analysis:** Lizah Nyawira, Anita Musiega, Sassy Molyneux, Benjamin Tsofa, Edwine Barasa.

**Funding acquisition:** Edwine Barasa.

**Investigation:** Edwine Barasa.

**Methodology:** Rahab Mbau, Edwine Barasa.

**Project administration:** Edwine Barasa.

**Resources:** Edwine Barasa.

**Supervision:** Edwine Barasa.

**Writing – original draft:** Lizah Nyawira, Edwine Barasa.

**Writing – review & editing:** Lizah Nyawira, Rahab Mbau, Julie Jemutai, Anita Musiega, Kara Hanson, Sassy Molyneux, Charles Normand, Benjamin Tsofa, Isabela Maina, Andrew Mulwa, Edwine Barasa.

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
