## [Decision Letter · Decision Letter 0]

18 Aug 2021

 PGPH-D-21-00180 Examining Health Sector Stakeholder Perceptions on the Efficiency of County Health Systems in Kenya PLOS Global Public Health

Dear Dr. Barasa,

Thank you for submitting your manuscript to PLOS Global Public Health. After careful consideration, we feel that it has merit but does not fully meet PLOS Global Public Health’s publication criteria as it currently stands. Therefore, we invite you to submit a revised version of the manuscript that addresses the points raised during the review process.

A rebuttal letter that responds to each point raised by the editor and reviewer(s). You should upload this letter as a separate file labeled 'Response to Reviewers'.

We look forward to receiving your revised manuscript.

Kind regards,

Dinesh Neupane

Academic Editor

Journal Requirements:

Reviewers' comments:

Reviewer's Responses to Questions 

**Comments to the Author**

1. Does this manuscript meet PLOS Global Public Health’s publication criteria? Is the manuscript technically sound, and do the data support the conclusions? The manuscript must describe methodologically and ethically rigorous research with conclusions that are appropriately drawn based on the data presented.

Reviewer #1: Yes

Reviewer #2: Yes

2. Has the statistical analysis been performed appropriately and rigorously?

Reviewer #1: N/A

Reviewer #2: Yes

3. Have the authors made all data underlying the findings in their manuscript fully available (please refer to the Data Availability Statement at the start of the manuscript PDF file)?

Reviewer #1: Yes

Reviewer #2: No

4. Is the manuscript presented in an intelligible fashion and written in standard English?

Reviewer #1: Yes

Reviewer #2: Yes

5. Review Comments to the Author

Reviewer #1: Dear Authors,

Thank you very much for your manuscript submission. The topic is relevant and interesting to me. However, there needs to be few improvements for further consideration which I am detailing below.

Introduction:

After line 56, create a paragraph briefly describing the health system structure & present health financing arrangements for Kenya. Also, include some of the government strategies for realization of UHC & progress towards those UHC goals for Kenya.

Line 58: Preferably, change 'Efficiency' to 'Health system efficiency and 'system' to 'health system'.

Lines 65-67, pg3, while structurally devolution led to two tiers of health system governance, kindly explain briefly the functional responsibilities of national & county health systems after devolution here.

Line 81-84: Kindly provide a reference explaining that tacit knowledge can be expressed through verbalization (FGD or interview as data collection tools) or writing (narratives). I would humbly suggest to replace 'tacit knowledge' with 'knowhow'. Kindly check the following books & one article if needed:

1. Polkinghorne, D. (2004). Practice and the human sciences: The case for a judgment-based practice of care. Suny Press. (pg 2-5)

2. Polanyi, M. The Tacit Dimension. 1966 Chicago.

3. Visser, F. S., Stappers, P. J., Van der Lugt, R., & Sanders, E. B. (2005). Contextmapping: experiences from practice. CoDesign, 1(2), 119-149.

Line 84: Preferably change 'empirical analysis' to 'quantitative assessment' here.

Methods:

Study Design & Data Collection:

Kindly consider improving this section with the below mentioned questions:

1. What is the study design and methodology (not just the data collection method)? Why such an approach has been selected to answer the research questions? (kindly use content from lines 533-549 here for the purpose)

2. What is your ontological & epistemological stance for this research? Some references which can be used:

(a) Giorgi, A. (2020). Reflections on certain qualitative and phenomenological psychological methods. University Professors Press.

(b)Holloway, I. (1997). Basic concepts for qualitative research. Wiley-Blackwell.

3. How did you attempt to ensure trustworthiness & rigour for your study? Preferably, use the below stated reference for criteria to answer this point: Lincoln, Y. S., & Guba, E. G. (1986). But is it rigorous? Trustworthiness and authenticity in naturalistic evaluation. New directions for program evaluation, 1986(30), 73-84.

4. Why did you choose Focus Group Discussion (FGD) for data collection?

5. Why specifically is the term 'workshop' being used instead of simply 'focus group discussion'?

Data Analysis:

Kindly break down the term 'we' here to show responsibilities of various researchers during this data analysis stage like you have explained for data collection in lines 117-119. This will ensure further transparency and enhance rigour. Further, any data triangulation, member checking, peer debriefing performed during or after data analysis?

Line 99-100: Kindly consider replacing these lines to "For this study, we adopted purposive sampling of participants to gain diverse perspectives from respondents in order to investigate our research objectives".

Kindly attach a copy of the group discussion topic guide as an annexure for this paper.

Line 124: Probably add a line: "We repeatedly listened to the audio recordings and checked with the content of the transcripts prior to data analysis".

Line 128: Modify "insights from the data" to preferably "insights from the anonymised transcribed data".

Ethics:

Mention here how data privacy & confidentiality was ensured.

Results:

Kindly start with a brief description of how your results section is laid out with a synopsis of the details of the expected themes that you are going to inform further.

Stakeholder Understanding of the Efficiency of County Health Systems:

Line 143: Kindly explain in 2-3 lines what is meant by health system inputs, outputs and outcome here for the convenience of the readers.

Line 145: Preferably modify to : "Participants highlighted that efficient healthcare delivery should be cost effective and responsive to community healthcare needs. Therefore, a system wide approach might be needed for improving health system efficiency."

Preferably place a line in between line no 157-158: "In some cases, the formal county health authority may not have practical authority for health system resource allocation & distribution in the country."

Relevant Inputs to the County Health System:

Line 172: Explain the relevance or importance of health system inputs in regulating health system efficiency here in 2-3 lines.

Line 173: Preferably modify : "Hard inputs were consistent with health system building blocks" to "Hard inputs were consistent with the health system strengthening (HSS) components".

Line 174: May be add health service delivery and health system governance explicitly here.

Line 175: Change "such" to "such as"

Lines 175-176: Improve the lines with the following additions: policies & guidelines for? managerial system for?

Between line 176-177 considering adding a line: " Participants highlighted the need for emphasis on soft inputs and also expressed the difficulty in assessing them to ascertain the health system efficiency."

Line 187: Consider replacing, "optimal mix of inputs for county" to "optimal mix of inputs especially healthcare commodities for county..."

Line 188: Preferably, change 'inputs' to 'health system strengthening components'.

Relevant Outputs of the County Health System

Kindly start with explaining the relevance or importance of health system output in regulating health system efficiency here in 2-3 lines.

Line 206-207: Preferably modify to: "Respondents felt that health output & outcome indicators used in quantitative assessment of county health system efficiency should represent the disease burden of the county."

Line 207-209: Preferably modify to: "They also highlighted the need to select standardized output & outcome indicators that could be implemented nationally across all counties of Kenya for facilitating valid comparisons."

Lines 209-213: Consider modifying to :"Participants highlighted the dual burden of disease in Kenya, with an increasing burden of NCDs along with a continued relevance for infectious diseases & RMNCAH. Therefore, respondents expressed the need for monitoring of intervention coverage and health outcome indicators relevant for both NCDs and infectious diseases which could be compared across 47 counties."

Line 224: consider changing "...the importance of including the quality of care provided by the county health system as an output." to "... the relevance of including quality of care provided at the healthcare institutions as an important dimension for assessing the health system efficiency."

Factors Affecting County Health System Efficiency

Public Finance Management

Lines 236-238: Is the disbursement issue between national & county level or from county to facility level? The excerpts from lines: 241-243: "Money was received by counties by 20th December [2018] and three to four months down the line, money was still not at facility level and these are facilities that are not collecting user fees." suggest county to facility level?

This is important as huge amount of health finances (roughly i guess 60%) goes to the county level in Kenya.

Is there any content from the transcript to dive deeper into the issue of why there is delay in disbursement? If not, then kindly show it a scope for future research later in the conclusion section of the paper.

Line 248: Preferably modify: "...autonomy of health facilities" to "autonomy of health facilities and the need to assess facility level efficiency". ( taken from lines 265-266)

Lines 249-251: Please mention pathways in which funds are collected by public health facilities ( only user fee or other options as well?).

Human Resources for Health

Preferably use the term "Health Professionals" or "Healthcare Staff" than "health workers" in appropriate places ( where you talk about specialty doctors etc.) .

Lines 273-275: Preferably modify to : "For example, participants felt that the efficiency of county health system was affected by high workload and inadequate number of health workers employed by counties. " ( info taken from lines 281-284)

Line 286: Change 'mix' to 'skill mix' and 'health workers' to 'health professionals'.

Lines 287-292: Preferably change to: " There was maldistribution of health professionals, especially medical specialists, across the counties. This was basically due to absence of any joint formal recruitment strategy and lack of cooperation for sharing health professionals between counties. Participants suggested a coordinated sector wide collaboration and sharing of resources for an improved health system efficiency. (taken from lines 302-304)"

Lines 304-308: From this excerpt, why a smooth mechanism was difficult might need more exploration. Is there any content to further go deeper on the bottlenecks preventing sharing of resources between counties. If there, then kindly enrich this section further with the details.

Probably suggest in the paper conclusion section that further study exploring factors influencing absenteeism of health workers/ professionals could be explored.

Line 315: Probably Add a line here: "Moreover, there are also governance issues where the officials do not consider healthcare staff as a priority, resulting in delayed payment of salaries." (taken from lines 321-323)

Line 325-328: Preferably modify to : "Participants also reported that inadequate accountability mechanisms for especially permanent & pensionable health professionals contributed to health system inefficiency."

Political interests & interference:

Line 343:Consider improving “...health budgets to capital assets and infrastructure over other forms of investments such as health commodities or health workers.” to “...health budgets to tangible capital assets and infrastructure over other forms of intangible investments such as health commodities or health workers.”

Probably add a line between 357-358: " This resulted in discouraging people from taking up the post of health manager leading to further health system inefficiencies."

Corruption: This section is allright.

Co-ordination between actors:

This section provides good information on the lack of coordination of actors and its implications (like duplication) leading to health system inefficiencies. Could this section be possibly further enriched using the transcribed data to understand some of the reasons for this lack of co-ordination.

Discussion:

Overall, it is a very well crafted discussion.

Line 445: Preferably replace "software aspects of health systems" to "the intangible health system factors".

Line 460-462: Kindly check "national government to county government" as the excerpts only focused on county to facility level government.

Line 465: Probably replace 'had lost' to 'lacked'.

Line 479-480: (in) is not clear.

Line 533-549: My opinion is that these lines are more suitable in study design to justify the qualitative study approach.

Limitations: Consider changing the heading to study limitation.

Very well written limitation section. Some considerations: Was there any selection bias with purposive sampling? How did the researchers prevent their understanding from influencing the study outcomes?

Write in line 561: "However, the availability of thick description of the study context with rich data excerpts might help in the transferability of study findings into similar care contexts. "

Conclusion:

Needs to improve. Write few lines on what the findings indicated as the concept of efficiency and which were the sources of health system inefficiencies.

General: Check ref. no. 7 in the references for typographic error. Just check with COREQ checklist if all items have been covered. I would suggest to include a section on researcher reflexivity and mention the relevant details there based on domain 1 of COREQ checklist.

Reviewer #2: General: It is a well-drafted paper using a qualitative method to study the understandings of health system efficiency and perception on associated factors among health sector stakeholders in the counties health system of Kenya. I found the paper interesting.

My remarks on the manuscript are below:

1. The authors should have introduced more on the concept of health system efficiency in the background or methods section. A brief on: How does this study look different from many other studies mostly looking at efficiency at the institutional level? Where do the widely discussed concepts on technical efficiency and allocative efficiency fit or stand out in this study?

2. In the abstract conclusion: can you please add more (public health system implications) beyond methodological recommendations.

3. In line number 107: ….backgrounds were selected as detailed in Table 1” I could not find the table attached as mentioned.

4. Could you please add or supplement the revised manuscript with the workshop or focus group discussion guidelines that you have used as the study tools?

5. In line number 207-209 “They also highlighted the need to select outputs and outcomes that were applicable across all counties in Kenya to facilitate valid comparisons.” Beyond this line the text does not discuss about outcomes. Could you please discuss and present what outcomes were highlighted by participants?

6. Shortening the verbatim could enhance the readability of the manuscript. Verbatims are longer.

7. In the result section in subsection PFM: Only the aspect of budget execution and disbursement are presented. So the further discussion on other aspects of PFM like the role of the county on budget formulation, external security, and audit that could affect the efficiency could enrich this manuscript.

8. In line number 530, Please correct the reference abiding by the Journal styles.

9. The paragraph between lines number 528-521 discussed about the importance of qualitative research methods could be summarized and shortened.

10. Please add and elaborate strengths of this study along with the limitations?

11. What are the implications of this study's findings on the public health system of Kenya? Please elaborate.

6. PLOS authors have the option to publish the peer review history of their article (what does this mean?). If published, this will include your full peer review and any attached files.

**Do you want your identity to be public for this peer review?** For information about this choice, including consent withdrawal, please see our Privacy Policy.

Reviewer #1: No

Reviewer #2: No

---

## [Editor Report · Decision Letter 1]

15 Nov 2021

Examining Health Sector Stakeholder Perceptions on the Efficiency of County Health Systems in Kenya

PGPH-D-21-00180R1

Dear Dr. Barasa,

We're pleased to inform you that your manuscript has been judged scientifically suitable for publication and will be formally accepted for publication once it meets all outstanding technical requirements.

Within one week, you'll receive an e-mail detailing the required amendments. When these have been addressed, you'll receive a formal acceptance letter and your manuscript will be scheduled for publication.

An invoice for payment will follow shortly after the formal acceptance. To ensure an efficient process, please log into Editorial Manager at https://www.editorialmanager.com/pgph/ click the 'Update My Information' link at the top of the page, and double check that your user information is up-to-date. If you have any billing related questions, please contact our Author Billing department directly at authorbilling@plos.org.

Kind regards,

Dinesh Neupane, Phd

Academic Editor